# Clinical Characterization and Outcomes of Culture- and Polymerase Chain Reaction-Negative Cases of Infectious Keratitis

**DOI:** 10.3390/diagnostics13152528

**Published:** 2023-07-29

**Authors:** Sarah Atta, Rohan Bir Singh, Keerthana Samanthapudi, Chandrashan Perera, Mahmoud Omar, Shannon Nayyar, Regis P. Kowalski, Vishal Jhanji

**Affiliations:** 1Department of Ophthalmology, University of Pittsburgh School of Medicine, Pittsburgh, PA 15213, USA; atta.sarah@medstudent.pitt.edu (S.A.); samanthapudi.keerthana@medstudent.pitt.edu (K.S.); mao91@pitt.edu (M.O.); nayyar.shannon@medstudent.pitt.edu (S.N.); kowalskirp@gmail.com (R.P.K.); 2Department of Ophthalmology, Massachusetts Eye and Ear, Harvard Medical School, Boston, MA 02115, USA; rohan_singh@meei.harvard.edu; 3Department of Ophthalmology, Leiden University Medical Center, 2333 ZA Leiden, The Netherlands; 4Department of Ophthalmology, Stanford University School of Medicine, Stanford, CA 94305, USA; chandrashan@gmail.com; 5The Charles T. Campbell Ophthalmic Microbiology Laboratory, University of Pittsburgh Medical Center, Pittsburgh, PA 15213, USA

**Keywords:** keratitis, bacterial keratitis, herpetic keratitis, culture-negative keratitis, PCR-negative keratitis

## Abstract

Purpose: To examine the clinical presentation, management, and outcomes of culture and polymerase chain reaction (PCR) negative cases of infectious keratitis. Methods: In this retrospective case series, we evaluated the laboratory and medical records of culture- and PCR-negative cases (2016–2020) reported to a tertiary care center, which were presumed to be infectious keratitis on the basis of clinical history and presentation. Results: A total of 121 cases with culture-negative keratitis were included in this study. The mean age of the patients was 48.42 ± 1.89 years, and 53.72% were female. At presentation, the presumed etiology was viral in 38.01%, bacterial in 27.27%, fungal in 8.26%, Acanthamoeba in 6.61%, and unlisted in 28.92% of cases. The most common risk factors were a previous history of ocular surface diseases (96.69%) and contact lens use (37.19%). In total, 61.98% of the patients were already on antimicrobial medication at presentation. The initial management was altered in 79 cases (65.29%) during the treatment course. Average presenting and final (post-treatment) visual acuities (VA) were 0.98 ± 0.04 (LogMAR) and 0.42 ± 0.03 (LogMAR), respectively. A significantly higher frequency of patients with a final VA worse than 20/40 (Snellen) had worse VA at initial presentation (*p* < 0.0001). A history of ocular surface disease, cold sores, and recurrent infection (*p* < 0.05) were more commonly associated with a presumed diagnosis of viral keratitis. The patients with presumed bacterial etiology were younger and had a history of poor contact lens hygiene (*p* < 0.05). Conclusions: We observed a distinct difference in clinical features among patients with culture-negative and PCR-negative keratitis managed for presumed viral and bacterial infections. Although there was significant variability in presentation and management duration in this cohort, the visual outcomes were generally favorable.

## 1. Introduction

Infectious keratitis is one of the leading causes of ocular morbidity and corneal blindness globally, with an incidence of 2.5 to 799 per 100,000 person-years [1,2,3]. The burden of infectious keratitis is significantly higher in developing countries due to ocular trauma associated with agricultural work compared to developed countries where contact lens (CL) wearers are at a higher risk [4,5,6]. The clinical diagnosis of keratitis primarily relies on detecting the causative organisms after microbiological evaluation. Therefore, it is often challenging for ophthalmologists to manage keratitis cases (~30–40%) in which no organism is isolated from the corneal scrapings or polymerase chain reaction (PCR) assessment [7].

The lack of causative organism detection in cases of culture- and PCR-negative cases is attributed to prior topical antibiotic therapy, fastidious organisms, slow pathogen recovery, or infection at an earlier stage in the disease course [7,8,9]. It has been postulated that these cases may be viral necrotizing stromal keratitis resembling infectious keratitis or comprise the inflammatory component of resolving corneal ulcers before scar formation [7]. It is essential to develop a clinical understanding of the associations between prior antibiotic treatment and culture settings, clinical presentation, and differential disease course to ensure that the patients in which no organism is isolated are not misdiagnosed, thereby preventing delay in the treatment and poor visual outcomes. The causative etiologies in these patients are based on the clinical suspicion of the treating ophthalmologist on the basis of clinical history and presentation.

In this study, we performed a retrospective analysis of the clinical presentation, associations, and management of patients with culture- and PCR-negative cases of keratitis who were presumed to have infectious etiologies and were treated at the University of Pittsburgh Eye Center over five years.

## 2. Methods

This single-center, retrospective study was approved by the Institutional Review Board and Ethics Committee of the University of Pittsburgh. The study was conducted in compliance with the Health Insurance Portability and Accountability Act (HIPAA) of 1996 and adhered to the tenants of the Declaration of Helsinki. We reviewed clinical charts of all patients who had microbiological culture and PCR reports at the Charles T. Campbell Microbiology Laboratory between January 2016 and December 2020. The patients with a diagnosis of keratitis who had no microbial culture growth or were PCR-negative were identified for chart review. The cases with inaccessible or incomplete patient charts or follow-ups with outside providers were excluded.

At our center, corneal and conjunctival scrapings are inoculated on 5% sheep-blood-supplemented agar, chocolate agar, and mannitol agar. Post-sampling, the scraped sample is either transferred directly to the culture plates or submitted to our laboratory in a liquid transport medium. The sample is inoculated in a “C streak” or in a grid pattern. In suspected cases of fungal keratitis, the sample is inoculated on Sabouraud’s dextrose agar with chloramphenicol or gentamycin, whereas, in cases of suspected Acanthamoeba keratitis, the sample is inoculated on a non-nutrient agar with an E.coli overlay. The media are incubated at room temperature and examined every day. Two blood agar samples are plated: one to be incubated in aerobic conditions and the other to be incubated in anaerobic conditions. If no growth is observed after 7 days on the blood and chocolate agar culture plates, or three weeks for non-nutrient agar, the plates are deemed negative for the organisms. The fungal culture plates are incubated at 26 ± 1 °C and examined daily for 2–3 weeks. If there is no growth on Sabouraud’s dextrose agar, the sample is considered negative for fungal infection. The standardized criteria for culture positivity are confluent growth at the inoculation site on solid-phase media; growth of the same microorganism on more than one solid-phase medium and consistency between culture and microscopy findings; and repeated isolation of the same microorganism after different scrapings [10].

PCR is a highly sensitive and rapid molecular method for detecting microorganisms by DNA amplification of organisms in small samples with minimal microbial load. The sample is obtained using the same method as performed for culture assays. The specimen is then placed in a lysis PCR buffer at a low temperature. Subsequently, DNA extraction is performed to separate DNA from the rest of the cellular components with standardized methods using commercially available kits. Following DNA extraction, a PCR mix is prepared by using a master mix. The reaction mixture consists of template DNA; the four nucleotides; two primers (bacterial, fungal, and viral specific), including one forward and one reverse; a buffer; and the Taq-DNA polymerase. The PCR primers are 18–25-base-pair-long single-stranded DNA samples that match the sample DNA, and they are the starting points for synthesis of the new strands. The primers are specific for different microorganisms or specific target sequences in microorganisms such as the 16S ribosomal ribonucleic acid (rRNA) gene and the 18S rRNA gene for bacteria and fungi, respectively. The mixture is then placed in a thermocycler, in which the cycles consisting of template denaturation, primer annealing, and new strand extension are performed repetitively. Each cycle is initiated by denaturation by rapidly increasing the temperature to 92–96 °C for 15–60 s to dissociate the two DNA strands. Subsequently, the mixture is rapidly cooled to a temperature of 42–75 °C for primer annealing. The complementary strand is synthesized by Taq polymerase by addition of ~100 nucleotides per second to complete the complementary strand at 72–74 °C. Each cycle is stopped by raising the temperature to 92 °C.

The documented data of the patients meeting the pre-set inclusion criteria included demographic information, suspected etiology, ocular and systemic risk factors, symptom duration before presentation, initial and final visual acuity, infiltrate size and location, presence of hypopyon, medical management, duration of treatment, and time to epithelial defect closure.

## 3. Statistical Analysis

Python (v3.7.0) was used for statistical analysis, specifically the scientific packages pandas (v3.6), NumPy (v1.18.5), and scipy (v1.4.1). Chi-squared and *t*-tests were used to calculate *p*-values for discrete and continuous variables. The data are presented as mean ± standard error of the mean. A *p*-value of less than 0.05 was considered statistically significant.

## 4. Results

The specimens were collected for 1694 keratitis cases and sent to the laboratory during the study period. Among these, we identified 398 (23.49%) culture- and PCR-negative cases per the pre-set criteria. However, upon chart review, we found that 184 cases were not diagnosed as infectious keratitis during follow-up evaluations, and 93 did not have accessible encounters in the medical records. A total of 121 cases of culture- and PCR-negative cases of infectious keratitis were recorded in the remaining laboratory reports and included in the comprehensive chart review.

Over half of the patients included in the cohort were female (53.72%, 65). The average age at presentation was 48.42 ± 1.89 years (range: 4 to 90 years) (Table 1). Based on the clinical presentation, 46 (38.01%) patients were diagnosed with presumed viral keratitis. The remaining cases had a presumptive diagnosis of bacterial (28.92%, 35), fungal (8.26%, 10), and Acanthamoeba (6.61%, 8) keratitis. A previous history of ocular surface disorders (such as dry eye disease, conjunctivitis, blepharitis, recurrent corneal erosions, meibomian gland disease, neurotrophic, recent corneal abrasion, keratopathy, rosacea, trauma, and anterior basement membrane dystrophy) (96.69%, 117), contact lens use (37.19%, 45), and corneal surgical procedures (such as anterior lamellar keratoplasty, penetrating keratoplasty, LASIK, superficial keratectomy, and endothelial keratoplasty) (26, 19.83%) were the most common risk factors among affected patients (Table 2). Amongst the CL users, 31 (25.61%) patients reported poor CL hygiene. In the cohort, 19 patients had a history of recurrent infections. Systemic risk factors included a history of autoimmune disease (14.04%, 17) and prednisolone and other systemic steroid use (37.19%, 45) (Appendix A).

Average visual acuity (VA) at presentation was 0.98 ± 0.04 (LogMAR), average intraocular pressure was 14.80 ± 0.49 mm Hg, and average epithelial defect size was 8.53 ± 1.26 mm^2^. Amongst the cohort, 18 (14.88%) patients had hypopyon, and 2 (1.6%) had hyphema. Five patients (4.13%) required admission after presentation for further management. The average duration of symptoms before presentation was 13.76 ± 1.93 days. An array of microbiological studies was performed at the presentation. Additional microbiological investigations were performed when necessary, including PCR for herpes simplex and zoster, viral culture, Acanthamoeba culture and PCR, Adenoviral PCR, and Chlamydia amplification. More than half of the patients (61.98%, 75) included in the study were already on antimicrobial medication prescribed by an outside provider.

Antimicrobial management was modified in non-responsive patients during the treatment course in 79 cases (65.29%), with 35 patients (28.93%) being switched to antiviral therapy. In the study cohort, 53 (43.80%) patients required additional treatment in the form of bandage contact lenses (27.27%, 33) and serum eye drops (25.62%, 31). Some of the cases required amniotic membrane placement (5.79%, 7), cyanoacrylate glue (1.65%, 2), tarsorrhaphy (3.30%, 4), penetrating keratoplasty (2.48%, 3), superficial keratectomy (0.83%, 1), and evisceration (0.83%, 1) (Table 3).

The average defect resolution time was 48.21 ± 4.07 days, the average treatment duration was 98.28 ± 121.85 days, and the average follow-up length was 202.34 ± 21.29 days. Treatment duration varied between the presumed etiology of culture-negative keratitis. Cases with viral etiologies averaged 113.63 ± 9.81 days for treatment, whereas cases with bacterial etiologies averaged 86.42 ± 7.36 days for treatment, and unknown etiologies averaged 81.23 ± 6.74 days for treatment. Similarly, follow-up time with an ophthalmologist for the management of culture- and PCR-negative keratitis also varied, with viral etiologies averaging 152.14 ± 15.44 days for follow-up, bacterial etiologies averaging 221.35 ± 29.42 days, and unknown etiologies averaging 253.55 ± 26.58 days.

The average final VA (post-treatment) was 0.42 ± 0.03 (LogMAR). Patients who had a final VA worse than 20/40 Snellen (0.301 LogMAR) were significantly older at presentation (*p* < 0.0001) and had worse VA at initial presentation (*p* < 0.0001) (Table 3). Longer symptom duration before presentation was significantly associated with worse visual outcomes (*p* = 0.0042). Other factors that were significantly associated with a worse final VA included a history of ocular surgery (*p* = 0.0003), topical corticosteroid use (*p* = 0.0173), recurrent infection (*p* = 0.0007), a history of keratitis (*p* = 0.0231), and a history of corneal procedures (*p* = 0.0148). A worse final VA was also associated with the need for adjunctive treatments (*p* < 0.0001), the use of serum drops in management (*p* < 0.0001), and a longer treatment duration (*p* = 0.014). The mean treatment duration for patients with better visual outcomes (final VA < 20/40) was significantly shorter (64.64 ± 2.08 days, *p* = 0.002) compared to patients with worse (148.8 ± 11.15 days) visual outcomes (final VA>20/40). Similarly, defect closure duration in patients with better visual outcomes was significantly shorter (30.97 ± 4.14 days, *p* < 0.0001) compared to patients with worse (71.85 ± 10.88 days) visual outcomes (Figure 1).

The average age of patients with presumed culture- and PCR-negative keratitis was significantly higher (53.28 ± 2.62 years) compared to the rest of the cohort (45.72 ± 2.41 years, *p* < 0.0001) (Table 4). A history of ocular surface disease was significantly associated with having presumed viral etiology for keratitis (*p* = 0.0014), including the history of infectious keratitis (*p* = 0.0055). Other common risk factors significantly associated with presumed viral etiology culture- and PCR-negative keratitis included a history of cold sores (*p* = 0.0073) and recurrent infection (*p* < 0.0001). Expectedly, a significantly higher percentage of the patients with presumed infectious keratitis cases responded to valacyclovir (54.35%, 25), acyclovir (23.91%, 11), and ganciclovir (15.21%, 6). Expectedly, very few patients responded to fortified antibiotics (10.87%, 5) and fluoroquinolone eye drops (56.52%, 26). The mean treatment duration for patients with presumed viral keratitis was significantly longer (149.28 ± 23.78 days, *p* < 0.0001) compared to patients with keratitis due to other presumed infectious etiologies (49.24 ± 6.24 days) (Figure 2). The defect closure duration in patients with presumed viral keratitis was significantly longer (68.25 ± 10.23 days, *p* = 0.012) compared to patients with other presumed infectious etiologies (31.39 ± 4.28 days).

Among patients with presumed bacterial etiology for presumed infectious keratitis, the average age at initial presentation was significantly lower than that of other patients (*p* = 0.0041) in the cohort (Table 4). These cases were also associated with poor contact lens hygiene (*p* = 0.0479). Expectedly, these patients responded well to fortified antibiotics (72.73%, 24). The defect closure duration in patients with presumed bacterial keratitis (29.54 ± 4.95 days, *p* = 0.0024) was significantly shorter compared to patients with other presumed infectious etiologies (52.37 ± 5.39 days). The mean treatment duration for patients with presumed bacterial keratitis was also significantly shorter (34.74 ± 3.64 days, *p* < 0.0001) compared to patients with keratitis due to other presumed infectious etiologies (111.23 ± 13.28 days) (Figure 3).

## 5. Discussion

This retrospective study consisting of 121 patients is one of the largest studies evaluating the demographic and clinical characteristics of patients with cases of presumed infectious keratitis. In this cohort, a history of ocular surface diseases (96.69%) and contact lens use (37.19%) were the most common risk factors. At presentation, the average visual acuity was 0.98 (LogMAR), and most of the patients were on antimicrobial medication, most commonly fluoroquinolones. The revised management approach included changing the prescribed antibiotics; almost half of the patients required adjunctive treatments such as bandage contact lenses. The average post-treatment visual acuity improved to 0.44 (LogMAR), with an average time to defect resolution of 48.21 days and an average treatment duration of 98.28 days. Worse visual outcomes were significantly associated with older age, worse VA at initial presentation, longer symptom duration before presentation, treatment duration, and the need for adjunctive treatment. The patients with presumed viral keratitis were older and had longer defect closure and treatment duration than the rest of the cohort. In contrast, the patients with presumed bacterial keratitis were younger and had shorter defect closure and treatment duration than the rest of the cohort. Additionally, patients presumed to have bacterial keratitis rarely had a history of previous corneal infections compared to patients who were presumptively diagnosed with viral keratitis.

In the literature, two studies have performed a comparative analysis of culture-positive and culture-negative keratitis cases, primarily including bacterial and fungal keratitis cases [7,11]. The patients in our study cohort had an average symptom of 13.76 days before presentation, which was comparable to the average duration of symptoms for culture-negative cases (13.2 days) reported in the study by Yarimada et al. but significantly shorter than that of cases reported by Bhadange et al. (27.6 days) [7,11]. The history of prior topical steroid use in cases in our study cohort was similar to that reported by Yarimada and colleagues (10.74% vs. 14.5%). However, a higher proportion of culture- and PCR-negative cases had a history of prior topical antibiotic use in our cohort (60%) compared to that reported by Yarimada and colleagues (46%) and a lower frequency than that reported by Bhadange and colleagues (77%). The corneal defect size in our cohort was significantly smaller than reported in a previous study (8.53 vs. 22.4 mm^2^) [7]. Additionally, the patients had better mean visual acuity at presentation and post-management (0.98 and 0.44, logMAR) compared to patients in studies by Yarimada and colleagues (1.91 and 1.27) and Bhadange and colleagues (2.57 and 2.34). It is noteworthy that both previous studies reported that the final visual acuity was comparable in both culture-positive and -negative cases of keratitis.

Several studies have evaluated the sensitivity of PCR for detecting microorganisms infecting corneal tissue, specifically in cases in which there was an inadequate number of specimens or a low organism load. Eleinen and colleagues assessed the PCR sensitivity in detecting bacterial and fungal organisms using 16S rRNA and 18S rRNA of the organisms, respectively [12]. They observed 87.88% and 90.91% of bacterial and fungal organisms compared to smears (bacterial: 33.33%; fungal: 65.91%) and cultures (bacterial: 57.33%; fungal: 59.09%). In a similar study, Badiee and colleagues reported a lower sensitivity (75%) on fungal PCR, whereas smear and culture sensitivity was reported to be 68% and 57%, respectively [13]. In a recent study, Shimizu et al. reported a comparative sensitivity of bacterial smear (63.1%) and PCR (63.6%), whereas sensitivity of culture was comparatively lower (51.8%) [14]. Zhao and colleagues assessed bacterial (5.8S rRNA), fungal (16S rRNA) and Acanthamoeba (conserved 29 regions of 18S rRNA and US4 region of envelope glycoprotein G) and reported a significantly higher sensitivity of 81.8% of PCR compared to 47.1% of cultures. The highly variable sensitivity of smear, culture, and PCR in different studies highlight the impact of technique on the sensitivity of these techniques [15].

The analysis of visual acuity data showed that patients who were prescribed topical steroids had worse visual outcomes. In a placebo-controlled randomized trial, topical prednisolone was reported to significantly improve visual outcomes in bacterial keratitis cases with baseline vision of counting fingers or worse, especially in patients with central corneal ulcers [16]. On the contrary, the indiscriminate use of topical steroids may further aggravate keratitis of unknown etiology; hence, it is suggested that steroid use should be avoided [17,18]. The effect of topical steroid use on visual outcomes may be attributed to a potential delay in re-epithelialization after topical steroid use [19]. We observed that the delay in epithelial defect closure was associated with worse visual outcomes, primarily due to more extensive scarring from the healing of a larger epithelial defect or due to the involvement of the stromal layer, which may delay healing [20]. Therefore, the time to epithelial defect resolution may be an observable measure that can be directly linked to final visual outcomes.

In this study, 38% and 27% of cases were presumptively diagnosed as viral and bacterial keratitis, respectively. In a previous study evaluating 1387 patients diagnosed with infectious keratitis with culture positivity, we had observed a higher percentage of bacterial (72%) and a lower percentage of viral (16%) agents [4]. The distinct difference in these data reflects that the majority of the cases with bacterial keratitis are diagnosed using established culture methodologies, whereas it is challenging to diagnose viral keratitis by microbiological assessment exclusively. In this study, the analysis showed that the patients presumed to have viral keratitis were approximately eight years older than those presumed to have a non-viral etiology. A similar observation was reported in a study evaluating cases of HSV keratitis, in which the incidence increased with age and was highest in those over 75 years old. In a rodent study, corneal HSV-1 infection was associated with worse pathology in adult mice than in younger counterparts without associated differences in tissue viral load, which was attributed to differences in the local immune response [21]. Furthermore, the history of ocular surface diseases and recurrent corneal infections could be an inherent predilection for viral keratitis to recur [22]. In presumed bacterial keratitis, cases caused by poor contact lens hygiene were observed to be present twice as often as cases caused by other presumed corneal infections. Bacterial keratitis is commonly associated with poor contact lens hygiene, specifically in developed countries [23,24,25]. The significantly shorter time for defect resolution observed in presumed bacterial cases is indicative of rapid bacterial clearance and inflammatory cessation, while viral cases may require a longer duration for viral load clearance, which targets older individuals who typically have sustained inflammatory responses.

In addition to the gestalt interpretation of physical findings, clinical presentation should guide the diagnosis based on risk factor association for common etiologies. In our study, patients with suspected bacterial etiology were younger, male, and had a history of poor CL hygiene and no previous history of corneal diseases. On the contrary, cases with suspected viral etiologies were more likely to be found in older patients with no history of poor CL hygiene but a history of ocular surface and corneal disease, cold sores, and recurrent infections. In addition, the clinical course in cases with a presumed bacterial etiology had a shorter follow-up time, treatment duration, and time to defect closure compared to cases with presumed viral etiology, with longer follow-up times, treatment duration, and time to defect closure. Interestingly, final visual acuity was comparable between the patients presumed to have bacterial (0.43 logMAR) and viral (0.44 logMAR) keratitis.

The presumptive viral keratitis cases were noted to have a variable management plan, typically including both antiviral therapy and antibiotics. Interestingly, 28.93% of these cases were initially started on broad-spectrum antibiotics, and their management plan was switched to antiviral therapy at a later stage, primarily due to non-responsiveness to the initial treatment regimen. Previous studies have postulated that culture negativity may be attributed to antimicrobial use before presentation, smaller defect size resulting in inadequate specimen collection, and the limited availability of appropriate culture types [7,8]. We also observed that the prior use of antimicrobial medication plays a role in culture negativity; however, other factors are also likely at play as approximately 35% of cases in our cohort did not have prior antibiotic use. Moreover, the sensitivity of viral cultures in the eye is lower than that of bacterial cultures [26]. Emerging technologies, such as next-generation sequencing and optical coherence tomography, may provide means for precise diagnoses, strengthening confidence that targeted therapies are implemented in cases of culture-negative keratitis [1,2].

The primary limitation of this study is incomplete reporting, which is inherent to retrospective studies and leads to imprecision and varying effect sizes. Furthermore, this study is limited by the percentage (35%) of patients lost to follow-up. This low follow-up rate may produce a bias towards estimating worse clinical outcomes than exist among patients with culture- and PCR-negative keratitis since patients who clinically improved more swiftly are less inclined to return. Finally, the study may not be widely generalizable as we are a tertiary care center, and more than 60% of patients referred to us are already on an antimicrobial, which may signify particularly intractable and resistant organisms and disease. Therefore, this may not represent culture- and PCR-negative keratitis in the community setting and is more likely to represent patients who could have had a positive culture if they had presented earlier.

In conclusion, the present study characterized one of the largest case series of culture-negative infectious keratitis with management guided by clinical suspicion. Viral etiology should be commonly suspected among culture- and PCR-negative keratitis cases; however, a younger age and poor contact lens hygiene were significantly associated with suspected bacterial etiology. There is significant variability in presentation and management duration depending on the type of etiology, with suspected viral etiologies requiring significantly longer follow-up time, treatment duration, and time to defect closure. Nevertheless, culture- and PCR-negative keratitis cases generally have favorable outcomes, with many patients achieving adequate sight preservation. Future studies on integrating emerging technologies such as next-generation sequencing may help improve the speed of the delivery of targeted therapies.

## Figures and Tables

**Figure 1 diagnostics-13-02528-f001:**
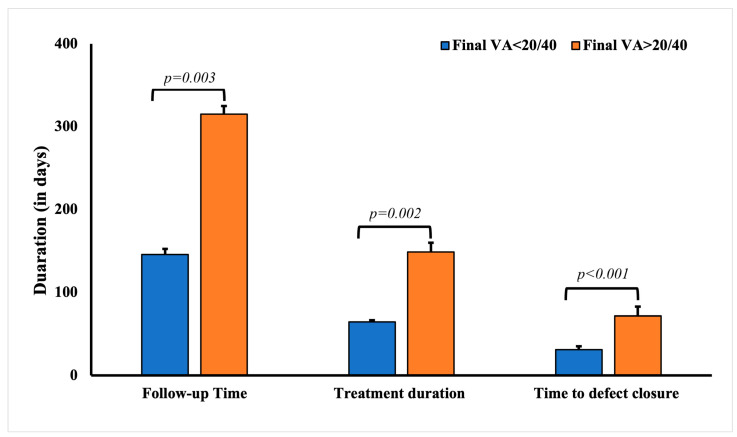
Comparative analysis of the follow-up, treatment, and defect closure duration in patients with culture-negative keratitis with final visual acuity <20/40 and >20/40.

**Figure 2 diagnostics-13-02528-f002:**
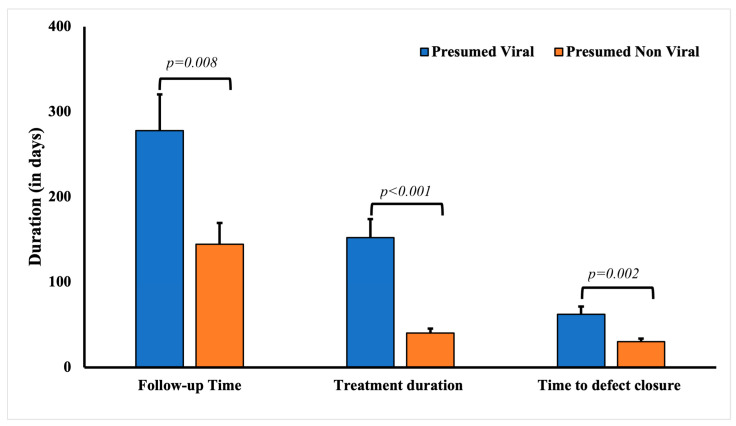
Comparative analysis of the follow-up, treatment, and defect closure duration in patients with culture-negative keratitis with presumed viral and non-viral keratitis.

**Figure 3 diagnostics-13-02528-f003:**
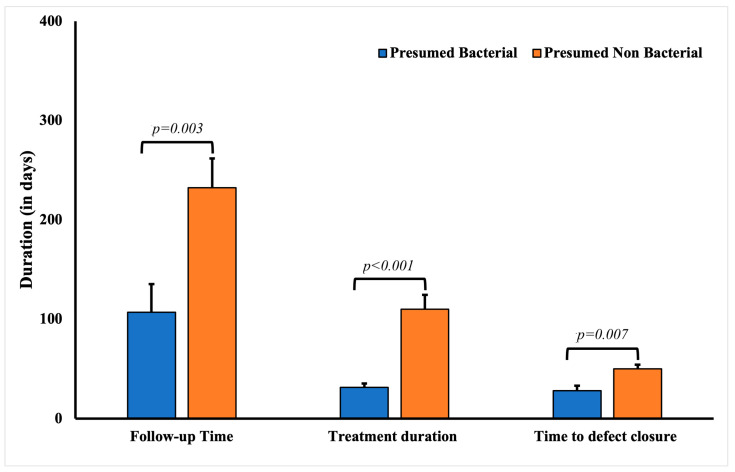
Comparative analysis of the follow-up, treatment, and defect closure duration in patients with culture-negative keratitis with presumed bacterial and non-bacterial keratitis.

**Table 1 diagnostics-13-02528-t001:** Demographics and presumed diagnosis of all the patients included in the study.

Demographics	*n* = 121	Percentage
Age at presentation (years)	48.42 ± 1.89	
Sex		
Female	65	53.72
Male	55	45.45
Unknown	1	0.83
Laterality		
Unilateral	111	91.74
Bilateral	10	8.26
Presumed diagnosis		
Viral	46	38.01
Bacterial	33	27.27
Fungal	10	8.26
Acanthamoeba	8	6.61
Unknown	35	28.92

**Table 2 diagnostics-13-02528-t002:** Clinical history and characteristics of patients included in the study.

	*n* = 121	Percentage
Defect Size (mm^2^)	8.53 ± 1.26	
Risk factors		
History of corneal surgical procedures	26	19.83
Contact lens use	45	37.19
Poor contact lens hygiene	31	25.61
History of infectious keratitis	3	2.48
Ocular surface disease ^†^	117	96.69
Recent water exposure	4	3.30
Ocular history		
Other ocular procedures ^‡^	26	21.48
Recurrent infection (other than keratitis)	16	13.22
Hypopyon	18	14.88
Glaucoma	14	11.57
Retinal Detachment	4	3.30
Others *	28	23.14

* Diabetic retinopathy, macular degeneration, uveitis, ptosis, ocular hypertension, dermatochalasis, amblyopia, hyphema, and epiretinal membrane. ^†^ Dry eye disease, conjunctivitis, blepharitis, recurrent corneal erosions, neurotrophic keratitis, recent corneal abrasion, and keratopathy. ^‡^ Cataract surgery, glaucoma procedure, panretinal photocoagulation, and retinal detachment repair.

**Table 3 diagnostics-13-02528-t003:** Significant characteristics among patients determining final visual acuity.

	Final VA Better Than 20/40 (*n* = 74)	Final VA Worse Than 20/40 (*n* = 47)	*p* Value
Age	42.27 ± 2.05	58.11 ± 3.16	<0.0001
Initial visual acuity	0.62 ± 0.09	1.58 ± 0.13	<0.0001
Topical corticosteroids	5.40% (4)	19.15% (9)	0.0173
History of infectious keratitis	0	6.38% (3)	0.0231
History of corneal surgical procedures ^‡^	13.51% (10)	31.91% (15)	0.0148
History of other ocular surgery *	12.16% (9)	40.42% (19)	0.0003
Recurrent infection	9.46% (7)	34.04% (16)	0.0007
Adjunct treatment	28.37% (21)	68.08% (32)	<0.0001
Serum eye drops	8.10% (6)	53.19% (25)	<0.0001
Bandage contact lens	18.91% (14)	40.42% (19)	0.0096
Amniotic membrane	1.35% (1)	12.77% (6)	0.0087

^‡^ Including penetrating keratoplasty, laser-assisted in situ keratomileusis, photorefractive keratectomy, deep anterior lamellar keratoplasty, and Descemet’s stripping endothelial keratoplasty. * Including cataract surgery, glaucoma procedure, panretinal photocoagulation, and retinal detachment repair.

**Table 4 diagnostics-13-02528-t004:** Characteristics of patients with presumed viral etiology at presentation.

	Presumed Viral (*n* = 46)	Presumed Non-Viral (*n* = 75)	*p* Value
Age at presentation	53.28 ± 2.62	45.72 ± 2.41	<0.0001
History of ocular surface disease ^†^	58.69% (27)	29.33% (22)	0.0014
History of infectious keratitis	28.26% (13)	13.33% (10)	0.0055
History of cold sores	13.04% (6)	1.33% (1)	0.0073
Recurrent infection	26.08% (12)	2.66% (2)	<0.0001
Response to valacyclovir	54.35% (25)	9.33% (7)	<0.0001
Response to acyclovir	23.91% (11)	1.33% (1)	<0.0001
Response to ganciclovir	15.21% (7)	2.66% (2)	0.0216
Response to fluoroquinolone drops	56.52% (26)	93.33% (70)	<0.0001
Response to fortified antibiotics	10.87% (5)	57.33% (43)	<0.0001

^†^ Including dry eye disease, lid laxity, conjunctivitis, blepharitis, recurrent corneal erosions, MGD, history of tarsorrhaphy, neurotrophic, lagophthalmos, recent corneal abrasion, keratoconus, keratopathy, ectropion, rosacea, trauma, ABMD, and Fuchs’ dystrophy.

## Data Availability

The research data are available from the corresponding author upon reasonable request.

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
