# Peer review of "Clinical Characterization and Outcomes of Culture- and Polymerase Chain Reaction-Negative Cases of Infectious Keratitis"

_diagnostics, 2023, doi:10.3390/diagnostics13152528_

Round 1

Reviewer 1 Report

very well written manuscript, objectives are clear and very nicely discussed.  

Author Response

We thank the reviewer for the positive feedback.

Reviewer 2 Report

Retrospective review of cases from one ocular lab of cornea samples that were negative for culture and PCR to determine risk factors.

Study would greatly benefit if data was compared with culture  and/ or PCR positive samples from the same lab.  Given current data set hard to determine any significant conclusions. 

Author Response

  1. Retrospective review of cases from one ocular lab of cornea samples that were negative for culture and PCR to determine risk factors.

Response: We thank the reviewer for their incisive feedback.

  1. Study would greatly benefit if data was compared with culture  and/ or PCR positive samples from the same lab.  Given current data set hard to determine any significant conclusions. 
    Response: We have provided a brief comparative analysis of culture positive keratitis cases reported in a previous study from our laboratory in the discussion section. [1]

The primary goal of this study is to highlight the demographic and clinical presentation in cases that are culture and PCR-negative cases with a likely infectious etiology, so that the treating ophthalmologists do not misdiagnose and delay the treatment in these patients.

[1] Kowalski, R.P.; Nayyar, S. V.; Romanowski, E.G.; Shanks, R.M.Q.; Mammen, A.; Dhaliwal, D.K.; Jhanji, V. The Prevalence of Bacteria, Fungi, Viruses, and Acanthamoeba From 3,004 Cases of Keratitis, Endophthalmitis, and Conjunctivitis. Eye Contact Lens 2020, 46, 265–268, doi:10.1097/ICL.0000000000000642.

Reviewer 3 Report

In this study authors analyze clinical features, management and outcomes of microbiological culture and PCR negative cases of infectious keratitis in a retrospective manner. They found significative differences in  clinical features, clinical presentation and management duration between suspicious cases of viral and bacterial infections. However, there are some observations that must be clarified:

In introduction and discussion sections there is no enough information about studies where results of microbiological culture and PCR test were analyzed. The results of this study could be compared with positive results and discuss about the differences.

In material and methods culture microbiological and PCR methodology must be described in detail, in order to understand the posible role of the technique in negative results. 

Author Response

  1. In this study authors analyze clinical features, management and outcomes of microbiological culture and PCR negative cases of infectious keratitis in a retrospective manner. They found significative differences in  clinical features, clinical presentation and management duration between suspicious cases of viral and bacterial infections. However, there are some observations that must be clarified:

Response: We thank the reviewer for their evaluation of our work and providing feedback about it.

  1. In introduction and discussion sections there is no enough information about studies where results of microbiological culture and PCR test were analyzed. The results of this study could be compared with positive results and discuss about the differences.
    Response: We have added a brief comparative analysis of the study conducted by our laboratory in the discussion section. [1]

[1] Kowalski, R.P.; Nayyar, S. V.; Romanowski, E.G.; Shanks, R.M.Q.; Mammen, A.; Dhaliwal, D.K.; Jhanji, V. The Prevalence of Bacteria, Fungi, Viruses, and Acanthamoeba From 3,004 Cases of Keratitis, Endophthalmitis, and Conjunctivitis. Eye Contact Lens 2020, 46, 265–268, doi:10.1097/ICL.0000000000000642.

  1. In material and methods culture microbiological and PCR methodology must be described in detail, in order to understand the posible role of the technique in negative results. 
    Response: We have provided a summary of the culture and PCR methodology in the method section of the manuscript.

Round 2

Reviewer 3 Report

In material and methods section the information required is not present in the new version of the manuscript. 

And in introduction and discussion sections more global information about studies where results of microbiological culture and PCR test were analyzed must be added  In order to discuss about the differences compared with the results of this study.

Author Response

Dear Dr. Kubica-Grygiel,

We thank you and the reviewers for the valuable feedback and for allowing us to resubmit the manuscript for consideration for publication in Diagnostics. We have carefully considered all the comments and revised the manuscript accordingly. The suggested revisions have helped in improving this manuscript. We have addressed each of the comments below in a point-by-point response (in italics) below:

In material and methods section the information required is not present in the new version of the manuscript. 

- We thank the reviewer for the insightful feedback. We have added the methods of culture and PCR in the methods sections.

And in introduction and discussion sections more global information about studies where results of microbiological culture and PCR test were analyzed must be added  In order to discuss about the differences compared with the results of this study.

- We have added 5 studies highlighting the sensitivity data of culture and PCR in infectious keratitis cases in the discussion section.

We would like to thank the editor and the reviewer for providing us with these constructive comments and hope that with the clarifications and revisions described herein our manuscript will now be suitable for publication.

Yours sincerely,

Vishal Jhanji, on behalf of all authors

Round 3

Reviewer 3 Report

The authors added the information required.